# Assessment of the Socio-Economic Impacts of Extreme Weather Events on the Coast of Southwest Europe during the Period 2009–2020

Rosa María Mateos [1,*], Roberto Sarro [1], Andrés Díez-Herrero [1], Cristina Reyes-Carmona [1], Juan López-Vinielles [1], Pablo Ezquerro [1], Mónica Martínez-Corbella [1], Guadalupe Bru [1], Juan Antonio Luque [1], Anna Barra [2], Pedro Martín [3], Agustín Millares [4], Miguel Ortega [4], Alejandro López [4], Jorge Pedro Galve [4], José Miguel Azañón [4], Susana Pereira [5], Pedro Pinto Santos [5], José Luís Zêzere [5], Eusébio Reis [5], Ricardo A. C. Garcia [5], Sérgio Cruz Oliveira [5], Arnaud Villatte [6], Anne Chanal [6], Muriel Gasc-Barbier [6] and Oriol Monserrat [2]

1 Department of Natural Hazards, Geological Survey of Spain (IGME), Ríos Rosas, 23, 28003 Madrid, Spain
2 Centre Tecnològic de Telecomunicacions de Catalunya (CTTC), Avinguda Carl Friedrich Gauss, 7, 08860 Castelldefels, Spain
3 (ASITEC) Ingeniería, Urbanismo y Medio Ambiente SL. Faisán, 21, 18014 Granada, Spain
4 Universidad de Granada, Avenida de la Fuente Nueva S/N C.P., 18071 Granada, Spain
5 Instituto de Geografia e Ordenamento do Território, R. Branca Edmée Marques, 1600-276 Lisbon, Portugal
6 Centre for Studies and Expertise on Risks, the Environment, Mobility and Urban Planning (CEREMA), Av. Albert Einstein, 13590 Aix-en-Provence, France
* Correspondence: rm.mateos@igme.es; Tel.: +34-958691021

**Abstract:** Coastal regions in Southwest Europe have experienced major interventions and transformations of the territory with unprecedented urban development, primarily related to growing tourism activity. The coast is the place where marine and terrestrial processes converge, making it highly vulnerable to the effects of climate change. However, the lack of information on the frequency of these extreme weather events and their impacts on the coast hampers an accurate analysis of the consequences of global change. This paper provides a detailed analysis of the extreme weather events (EWE) that have affected the Atlantic and Mediterranean coasts of Southwest Europe during the period from 1 January 2009 to 28 February 2020, as well as a quantification of their impacts: fatalities, injuries and economic damage. Official sources from France, Portugal and Spain were consulted, along with technical reports, scientific articles, etc., to generate a unified database. A total of 95 significant extreme events have caused 168 fatalities, 137 injuries and almost €4000 M in direct economic losses. Cyclone Xynthia (February 2010) on the French Atlantic coast stands out, having caused 47 fatalities, 79 injuries and substantial economic losses valued at €3000 M. The study shows a slight upward trend in the number of events recorded, especially during the last three years of the analysis, as well as in human losses and damages. The results reveal a higher exposure of the Mediterranean coast of Southwest Europe when compared to the Atlantic, especially the Spanish Mediterranean coast, with 61% of the fatalities recorded there during the study period. This is primarily due to a model of exponential tourism growth on the Mediterranean coast, with an enormous urban and infrastructure development during the last decades. Traditionally, the Mediterranean coast is less prepared to reduce the effects of marine storms, extreme events that are becoming more frequent and virulent in the context of climate and global change. This work highlights the need to create a continuous monitoring system–at the European level–of the impacts of extreme weather events on the coast, where 40% of the European population is concentrated. This observatory should serve as a source of information for risk mitigation policies (predictive, preventive and corrective), as well as for emergency management during disasters.

**Keywords:** extreme weather events; socio-economic impact; coastal regions; southwest Europe; climate change; geohazards management

## 1. Introduction

The EU has around 68,000 km of coastline which borders six main maritime basins: the Baltic Sea, the North Sea, the Northeast Atlantic Ocean, the Mediterranean Sea, the Black Sea and the outermost regions. Coastal areas are of vital importance not only because they are home to unique ecosystems, but also because they are preferential human settlement sites of enormous relevance to the European economy. Approximately 40% (around 206 million people) of the EU's population lives within 50 km of the sea and almost 40% of the EU's GDP is generated in these coastal regions [1]. The Mediterranean and the Northeast Atlantic Ocean coastlines are the most populated (36% and 29.5% of the EU coastal population, respectively). The projection for 2035 in Europe foresees an increase of 1.15% in the human population living in coastal areas [2].

Additionally, these coastal areas concentrate key infrastructures for world maritime trade (i.e., the ports of Barcelona, Valencia and Lisbon), strategic energy facilities (refineries, petrochemical and nuclear power plants), the fish farming industry (the Ebro delta, Galician estuaries, and French coast) and, of course, tourist resources associated with beaches and marinas (Costa del Sol, Costa Brava, and the Balearic Islands) and nature (wetlands such as Doñana and the Cíes Islands).

The enormous tourist development of these coastal regions has led to major interventions and transformations of the territory due to unprecedented urban planning. In many cases, the coastline is subjected to pressure far beyond its capacity, with serious environmental and social consequences. The case of the Spanish coastline is particularly significant. According to a recent report [3], over the last 30 years, the urbanized area of the Spanish coast has doubled. 13.1% of the coastline and 37% of Spanish beaches are urbanized. The magnitude in some areas is such that the coastline of Costa del Sol municipalities (South of Spain) is 90% urbanized.

Extreme weather events (EWE) have a severe impact on coastal areas and these regions are especially vulnerable to the effects of climate change. These extreme events can be defined based on the extraordinary nature of the data on temperature, precipitation, wind speed, sea level and atmospheric pressure, among other factors, including sea storms that can reach the level of hurricanes. They are also defined by their impact on society, including a high number of fatalities and high economic and environmental losses. Extreme weather events on the coast often lead to cascading processes: sea level rise, marine encroachment and coastal flooding, river flooding, coastal erosion, beach retreat, rockfalls and landslides in cliff areas, etc. The effects of the sum of these processes on an anthropized coastline can be devastating, causing fatalities and injuries; damage to housing, services, infrastructure and urban furniture, cultivated areas, and ecosystems; dragging of vehicles; interruption of activity, water and electricity supply cuts, etc.

Lozano et al. (2004) [4] examined the evolution in the occurrence of storms on the European Atlantic coast (from Ireland to Spain) for the period 1940–1998. Results indicate a seasonal shift in the wind climate, with more severe winters and calmer summers established regionally. This pattern appears to be linked to a northward displacement in the main North Atlantic cyclone track. On the other hand, during the last decades, several parts of the West Mediterranean basin have experienced significant changes in wave storm intensity related to an increase in wave storm durations. These relevant increases are primarily observed in the Alboran and Balearic basins [5].

According to recent studies [6–9] during the last decades, damage caused by extreme weather events in Europe has increased considerably, especially that related to heavy precipitation events and storm surges. The primary reasons are not only linked to climate change, but also to increased exposure of the population and other vulnerable elements. However, the lack of information on the frequency of these extreme weather events and their impacts on the coast hampers an accurate analysis of the effects of climate change and complicates the issue of objective criteria for the adoption of measures and the analysis of their effectiveness.

So far, the most frequent compilations of socio-economic damage in the scientific-technical literature refer only to a specific stretch of coastal territory (e.g., Barcelona metropolitan area [10]); to a specific economic sectors (e.g., damage to port infrastructures, [11]) to a type of impact (e.g., fatalities; [12–14]); to a disaster typology (e.g., floods; [15]); or are restricted only to a specific extreme event (e.g., storm Gloria; [16]). But there are few studies that address the estimation and analysis of losses over large territories, covering the coasts of several countries [17], and also including cascading events over a sufficient period to include dozens of extreme events. Only with such comprehensive quantitative assessments can we have an overview of the spatial and temporal trends associated with changes in triggering factors, exposure and vulnerability in order to draw useful conclusions for future risk management and mitigation measures.

This paper contains a detailed analysis of the extreme weather events that have affected the Atlantic and Mediterranean coasts of Southwest Europe during the period from 1 January 2009 to 28 February 2020 (before the COVID-19 pandemic), as well as a quantification of the impacts caused, including fatalities, injuries and economic damage. The study contributes to the research of the impacts of extreme events on the coast and to determine whether there is an upward trend in both frequency and damage, as suggested by IPCC models [18]. The study area encompasses 20 regions in south-western Europe: 3 in southern France, 5 in mainland Portugal and 12 in Spain (mainland and Balearic Islands). A distinction is made between the Atlantic and Mediterranean coasts with the additional aim of comparing the two coasts and analysing the strengths and weaknesses of each in the face of this type of process.

## 2. Study Area

The study area of the present work covers 20 regions of south-western Europe (Figure 1), 12 of the regions located on the Atlantic coast and the rest (8 regions) on the Mediterranean coast, distributed by country as follows:

- France (3 regions): Poitou-Charentes, Aquitaine and Languedoc-Roussillon.
- Portugal (5 regions): North, Central Region, Lisbon, Alentejo and Algarve.
- Spain (12 regions): Basque Country, Cantabria, Asturias, Galicia, Andalusian Atlantic Coast, Catalonia, Community of Valencia, Region of Murcia, Balearic Islands, Ceuta, Melilla and Andalusian Mediterranean Coast.

The study area covers 8034 km of coastline (864 km in France, 848 km in Portugal and 6322 km in Spain). Of the coastline analysed, 47.3% corresponds to the Atlantic coast (3807 km) and 52.7% to the Mediterranean coast (4227 km). The total area includes a population of nearly 42 million inhabitants (2021 census), with such important cities (in terms of population and socio-economic activity) as Bordeaux, Poitiers and Montpellier (France), Lisbon and Porto (Portugal) and Barcelona, Bilbao, Valencia and Malaga (Spain).

There are important differences between the Atlantic and Mediterranean coasts of Southwest Europe. The Atlantic coast is controlled by the powerful forces of tide, wind and waves which have formed a wide range of environment types; wind swept cliffs, exposed rocky headlands and narrow tidal inlets contrast sharply with long stretches of sandy beaches, sheltered bays and extensive intertidal mudflats. The most important rivers of the study area (Loire, Garonne, Douro, Tejo and Guadiana) drain into the sea along the Atlantic coast. The region benefits from an oceanic climate with mild, wet winters, cool summers and predominantly westerly winds. It is a coastline that is undergoing progressive anthropization with increasingly large urban and industrial areas (fisheries are strategic for the regional economy). This puts massive pressure on the natural environments, which include ecosystems of the highest interest, such as the Doñana National Park (Southern Spain), the largest wetland in Europe.

The Mediterranean coast of Southwest Europe has a very step relief due to the proximity of large mountain systems (the Pyrenees and the Baetic Mountains). The landscape includes high mountains and rocky shores, coastal wetlands and sandy beaches. Tides and swell are not as energetic as in the Atlantic, although during the Gloria storm (January 2020)

waves of 8.44 m in height were recorded on the coast of Valencia, the record measured in the Mediterranean so far [19]. The climate is characterised by hot, dry summers and cool winters with frequent heavy, short rainfall events. This Mediterranean region is one of the world's primary tourism destinations. The coastline is highly urbanised and has been severely modified. Additionally, there are chronic water shortages and a constant threat of forest fires. The extremely high tourist pressure has threatened enclaves of high ecological value, with the Manga del Mar Menor (Murcia region), the Albufera of Valencia and the Ebro Delta (Catalonia) standing out.

The economic importance of both coasts derived from tourism must be highlighted in the study area. As significant examples, we highlight the Balearic Islands in Spain, where tourist activity represents 45.5% of the region's GDP; a territory that received 16.5 million tourists in 2019. The Algarve coast in Portugal also represents a growing tourist destination, with slightly more than 9 million tourists for the same year, as well as the Languedoc-Roussillon coastline, with close to 1 million visitors in 2019 [20].

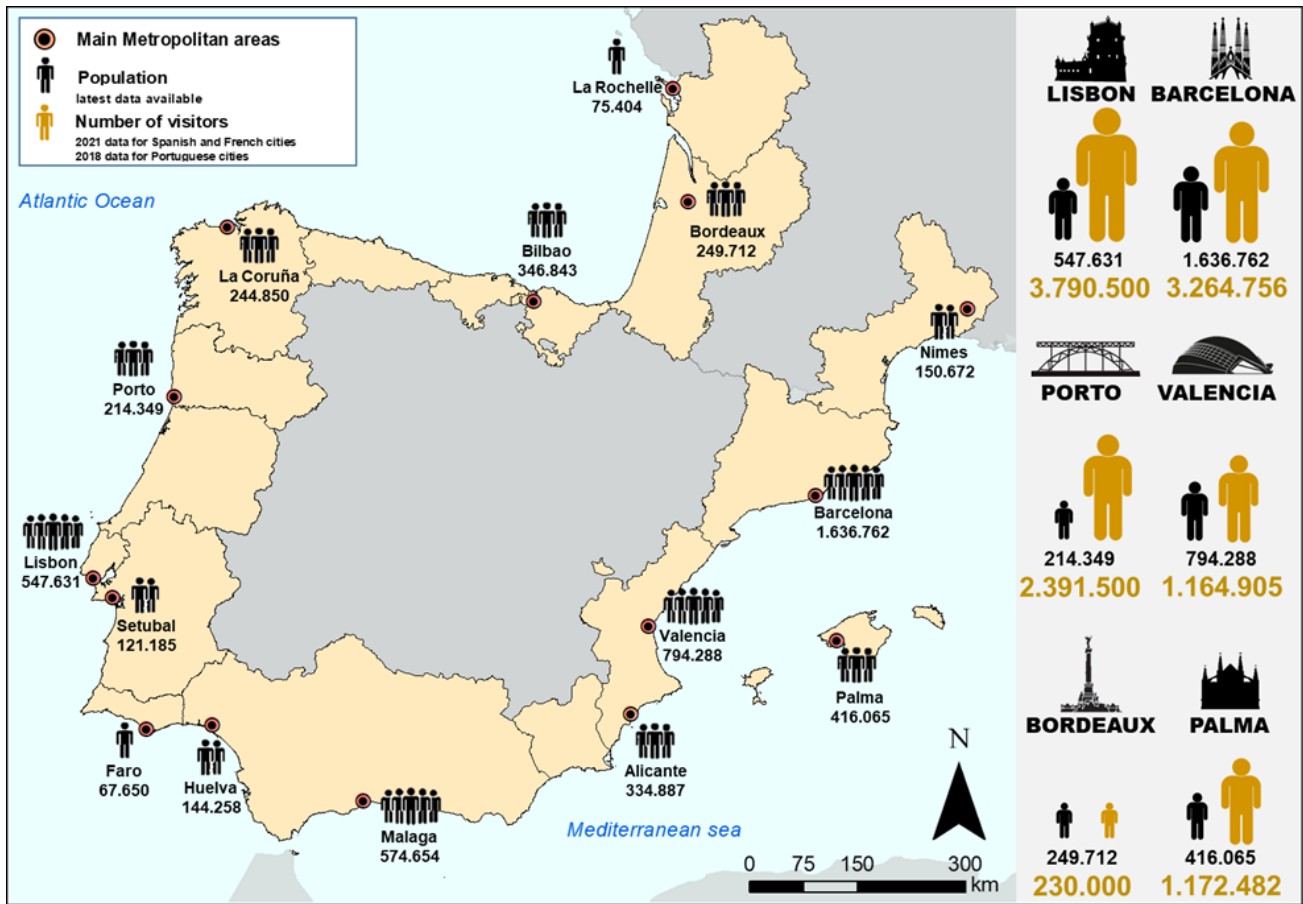

**Figure 1.** The 20 coastal regions of southwest Europe analysed in this paper, including 3807 km of Atlantic coastline and 4227 km of Mediterranean coastline. The most populated cities and the annual number of visitors for each are indicated. Data from [21–26].

## 3. Methodology

The study methodology consists of the following phases:

(1) A compilation of extreme weather events occurring on the coast during the period from 1 January 2009 to 28 February 2020. A preliminary database is created with the following fields:

- Identification of the event: event number, country, region, local name, starting date, final date, and coordinates of the primary focus.

- Characterization of the event: processes that took place (flooding, landslide, rockfall, beach retreat, coastal erosion, etc.), triggering factors, fatalities, injures, damage to, economic losses, emergency plan activated, actors involved, adopted measures and repairing costs.
- Source of information: official sources, technical reports, scientific articles, media, links of interest and photos attachment.

(2) Selection of the main extreme weather events using the following three criteria:

- A selection of different types of events was made by considering meteorologically significant events with a regional impact: Atypical Cyclonic Storm (TCA), High Level Isolated Depression (HLID), significant squalls (with their own name) and intense precipitation episodes (>50 mm/24 h) with a regional extension. Extreme sea level events are also included. The main sources of data are the repository of climatological summaries of the meteorological agencies of each country: Météo France (France); Instituto Português do Mar e da Atmosfera (Portugal) and Agencia Estatal de Meteorología (Spain). With this information, the database was completed by including relevant information for each event: duration and geographical extent, precipitation, temperature and precipitation anomalies, insolation and wind speed data. Some data concerning the maximum recorded wave height have been obtained from other national agencies related to port management. It should be noted that some events were supra-regional and international in scope, like the Gloria storm in January 2020, that hit many regions in the east of the Iberian Peninsula, as well as the Southeast coast of France.
- A selection of events with at least one fatality or injury due to extreme weather events on the coastline for the same period of analysis. In the case of Spain, official data were consulted from the repository of yearbooks and statistics of the Ministry of the Interior, that provides data on fatalities due to natural hazards recorded by the Directorate General of Civil Protection and Emergencies (DGPCE from its Spanish initials), as well as official reports from similar institutions, Sécurité Civile and Emergência e Proteção Civil, for France and Portugal, respectively. An important piece of information we have tried to collect is the cause of deaths and injuries and their relation to the type of process: flooding, being washed away by water currents, onslaught of waves, landslide or rockfall, collapse of structures, etc.
- A selection of events with economic damage of more than €0.5 M. For this purpose, official data from the Consorcio de Compensación de Seguros (Insurance Compensation Consortium of Spain) and official information from other institutions in France and Portugal have been used.

Additionally, the information in the three sections has been supplemented with other databases, scientific articles and technical reports. In the case of information gaps, contrasted data from the press and other media were consulted, to validate the estimates with the criterion that the same item of data appears as it is more than once.

Once the three main data sources (extreme weather events, economic damage and fatalities/injuries) were organised, they were overlaid to select the events with the greatest impact. Figure 2 shows the methodological phases used in the present work. The databases created for each country and ordered by dates are uploaded as "Supplementary files". These include the source data for each event.

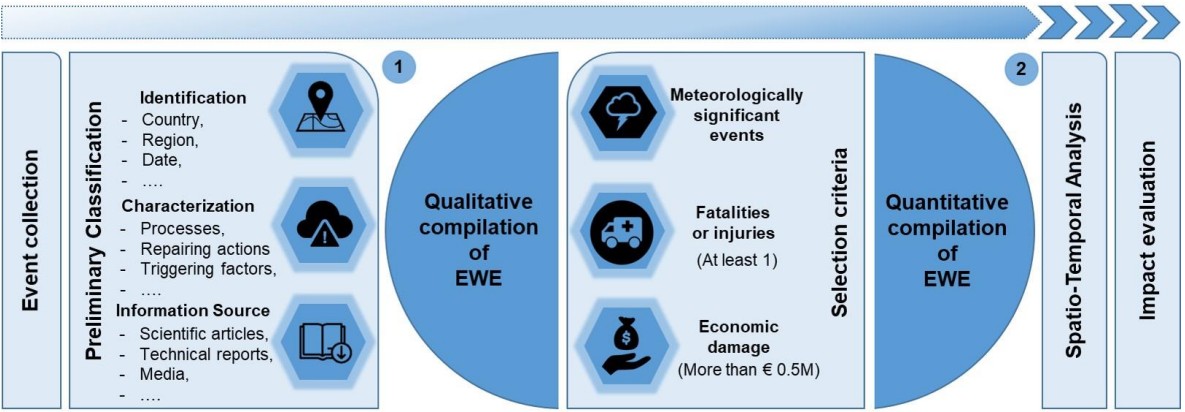

**Figure 2.** Methodology used for the present work with the different stages.

## 4. Results

The results of this work are presented by carrying out a spatial and temporal analysis of the data obtained. Finally, the main causes of death of the recorded fatalities are analysed.

### 4.1. Spatial Analysis

Based on the proposed methodology, the results of the number of extreme weather events, economic losses and fatalities for each of the countries are shown in Table 1. If we analyse the data by basin (Atlantic or Mediterranean), the results are shown in Table 2.

**Table 1.** Summary by country of the number of extreme weather events recorded, fatalities, injuries and economic damage caused during the period analysed.

| Country | Length (km) | Number of EWE | Human Impact | | Economic Losses (€M) |
| --- | --- | --- | --- | --- | --- |
| | | | Fatalities | Injuries | |
| France | 864 | 4 | 50 | 79 | 3113.9 |
| Portugal | 848 | 17 | 7 | 14 | 8.38 |
| Spain | 6322 | 74 | 111 | 44 | 827.9 |
| **Total** | **8034** | **95** | **168** | **137** | **3950.2** |

**Table 2.** Summary by basin (Atlantic and Mediterranean) of the number of extreme weather events recorded, fatalities, injuries and economic damage caused during the period analysed.

| Basin | Length (km) | Number of EWE | Human Impact | | Economic Losses (€M) |
| --- | --- | --- | --- | --- | --- |
| | | | Fatalities | Injuries | |
| Atlantic | 3807 | 49 | 65 | 93 | 3043 |
| Mediterranean | 4227 | 46 | 103 | 44 | 907.2 |
| **Total** | **8034** | **95** | **168** | **137** | **3950.2** |

As Table 2 shows, the distribution of extreme weather events by country is clearly unequal: Spain records 77.8% of the events, compared to 17.9% in mainland Portugal and 4.3% in the south of France. Table 2 shows a higher number of events in the Atlantic basin than in the Mediterranean one, despite the shorter coastline. The Mediterranean basin has a higher number of fatalities with 61.3% of the victims; this is in contrast to the number of injured persons, where the Atlantic basin has 67.8% of these.

Figure 3 represents the number of events recorded in each of the regions of the study area. We can observe a greater concentration in the Spanish Mediterranean strip, particularly in the region of Andalusia (South Spain), where 28 extreme weather events have left significant after-effects.

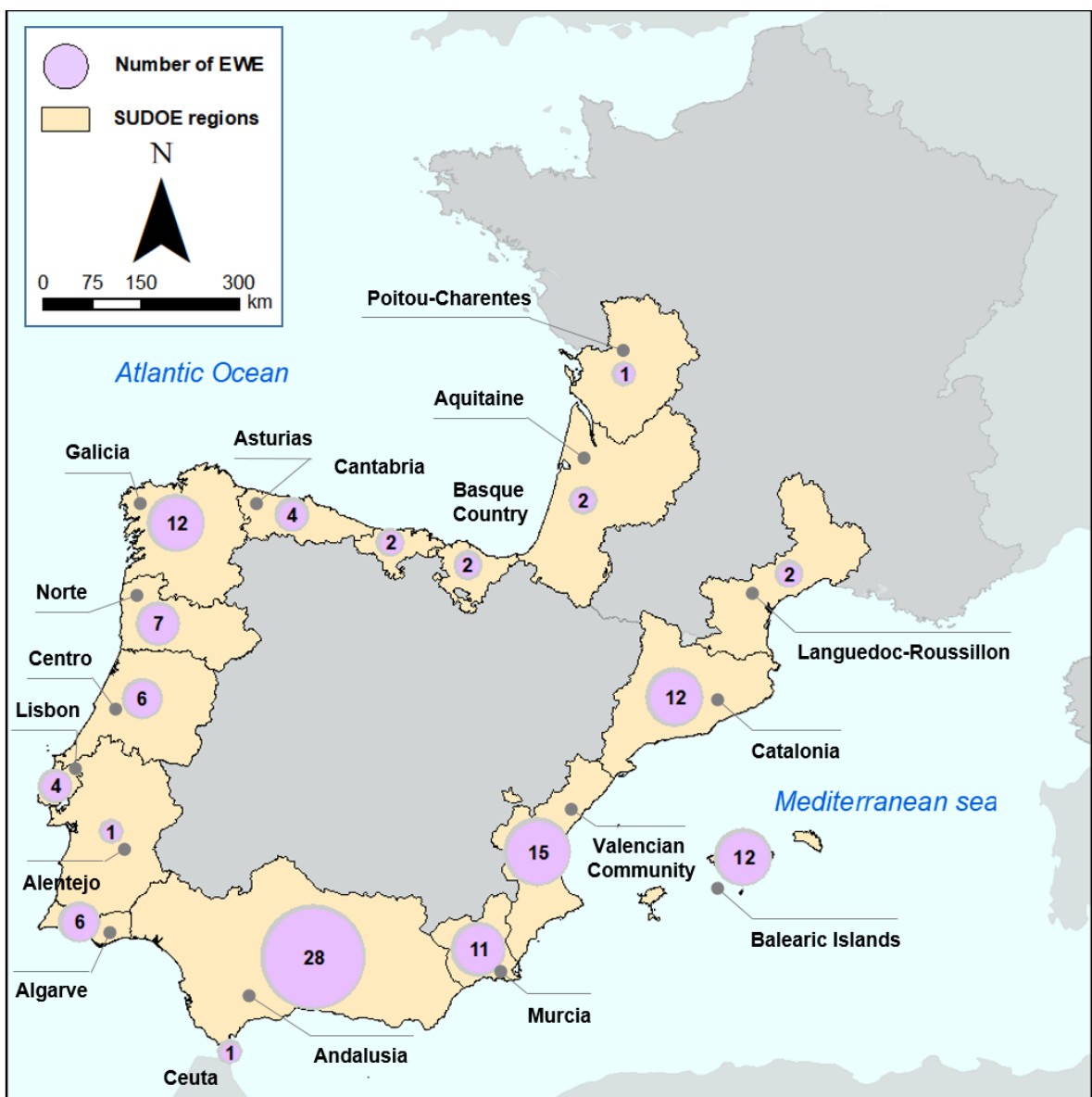

**Figure 3.** Representation of the number of extreme weather events in each of the unit-regions analysed. The Spanish Mediterranean coast, particularly the region of Andalusia, stands out with the highest number of events recorded. Note that the sum of the number of events in this figure exceeds the 95 inventoried, as some of them have an interregional dimension and are counted more than once.

Below is an analysis by country, identifying the most catastrophic events in each country.

### 4.1.1. France

France recorded 4 significant events in its three study regions: 2 in the Mediterranean region of Languedoc-Roussillon and 2 in the Aquitaine region. A total of 50 fatalities and 79 injuries were recorded for the period analysed, with economic losses quantified at €3113.9 M.

Cyclone Xynthia, that occurred on 27–28 February 2010, was undoubtedly the most tragic event of the last decade, not only in France, but in the whole of the territory analysed. Xynthia had an explosive development and a rapid displacement, bringing a strong flow of warm air over all continental Europe. On the surface, Xynthia was mainly a complex low until its landfall in the early hours of the 28th on the French Atlantic coast. Figure 4 shows the extent of the effects of the storm on the Atlantic coast of France under study, with a

greater virulence in the Gulf of Gascogne. The storm was characterised by heavy rainfall, strong winds (between 120 and 140 km/h), sea level rise and exceptional tides, which triggered a series of cascading processes [27]: extensive flooding, marine encroachment, channel overflows, mudflows, coastal erosion and landslides and rockfalls.

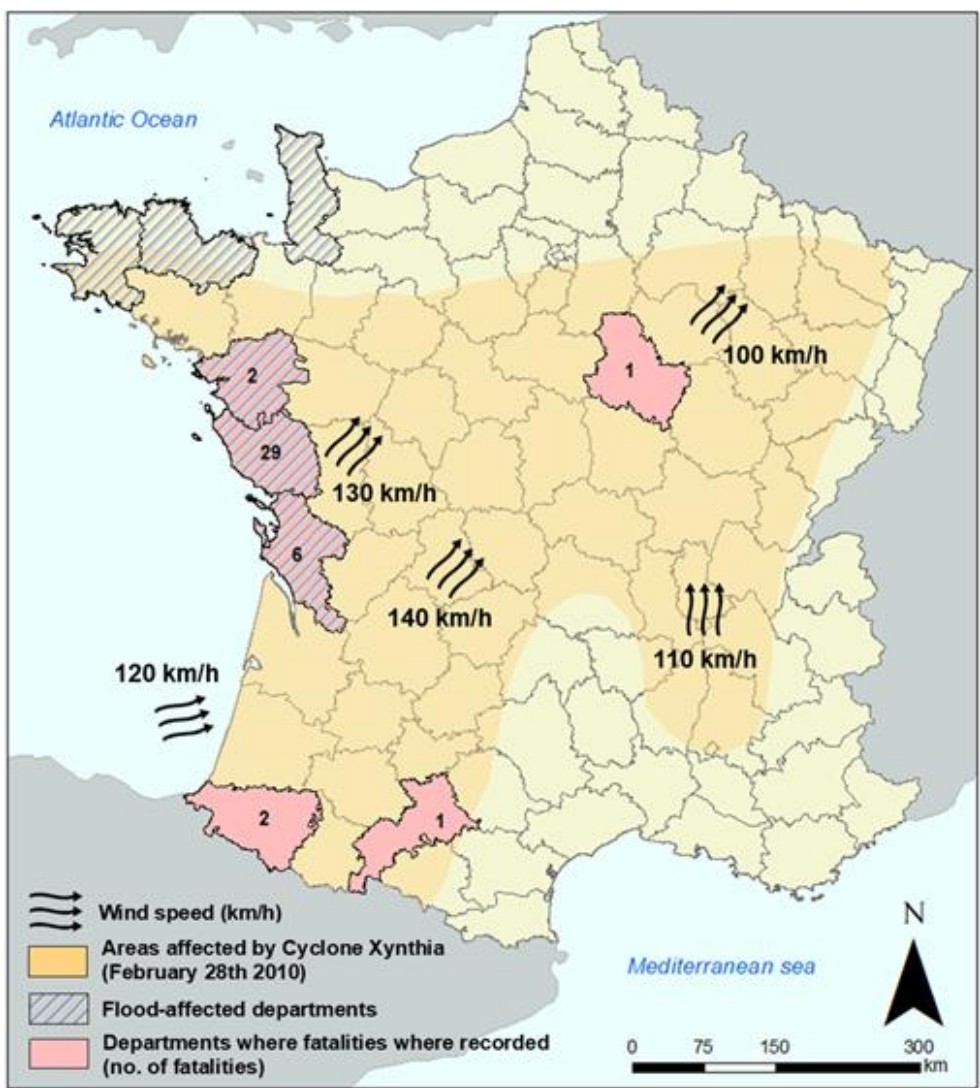

**Figure 4.** Impact of cyclone Xynthia (February 2010) on the Atlantic coast of southern France and the areas affected by the storm. Fatalities are indicated by departments. Data from [28,29].

The result of Xynthia was devastating: a significant amount of land (>50,000 ha) was flooded and thousands of homes (more than 5000 homes were damaged in Charente-Maritime), businesses and industrial areas were invaded with marine water. Numerous roads were affected as along with a multitude of vehicles. Losses in agriculture were heavy, with more than 52,000 hectares flooded in Charente-Maritime and Vendée. On the coast, in addition to port damage, numerous dykes were destroyed (more than 120 km of dykes were seriously affected in the Charente-Maritime region), and major erosion and shoreline retreat occurred, with significant loss of beaches.

The death toll from Cyclone Xynthia was a tragic one: 47 deaths and 79 injured. Community emergency plans were activated and local, regional and national authorities intervened. More than 2000 firefighters, 565 police officers and 330 civil protection personnel were involved. What followed was the evacuation and relocation of 767 people in the Vendée and approximately 2000 residents in Charente-Maritime; 1527 towns were declared disaster zones (Vendée, Charente-Maritime, Deux-Sèvres and Vienne). France activated

the Dike Plan and the Flash Flood Plan, as well as the Natural Hazards Prevention Plan. The economic costs generated by Cyclone Xynthia were substantial, estimated at around €3000 M [30].

### 4.1.2. Portugal

Portugal recorded 17 extreme weather events in the period studied, resulting in 7 fatalities and 14 injuries, with economic losses quantified at €8.38 M. It should be noted that reliable economic loss data is only available for three events, leaving the remaining 14 events unquantified, so the data shown in Table 1 for Portugal is certainly underestimated.

The Algarve coast has been the most affected in Portugal over the last decade. The most important processes on the Algarve coast are rockfalls on beaches with high exposure [31]. At Praia Maria Luísa (Albufeira), the most tragic event of the decade occurred on 21 August 2009. A rockfall of several tonnes on the beach cliff caused five fatalities and two serious injuries. The families of the victims filed a lawsuit seeking compensation of around €1 M.

However, the most extensive event and the greatest economic losses recorded took place between 30 October and 1 November 2011 on the North and Central coast of Portugal. It was a meteorological event of short duration, with heavy rainfall, strong winds, intense waves and unusual tides, that generated cascading processes: landslides and rockfalls, abnormal sediment accumulation, flooding, marine encroachment, coastal erosion and beach and cliff retreat. The coastal storm damage also affected built up areas, roads and agricultural land. After the storm, €500 M was invested in coastal protection measures, especially in the areas of Esmoriz, Cortegaça, Furadouro and Ovar.

### 4.1.3. Spain

The Spanish coastline was affected by 74 extreme weather events during the review period, which represents almost 78% of the total number of events recorded in the study area. The Mediterranean coastline stands out, with 44 events inventoried and accounting for almost 61% (102 victims) of the total number of fatalities recorded in the area of analysis (168). Economic losses amount to €827.9 M, although no reliable information is available for 30% of the events, so the figure is certainly underestimated.

Most of the events recorded on the Spanish Mediterranean coast took place during the early autumn after the intense heat of the summer, giving rise to HLIDs (High Level Isolated Depressions). The heat and humidity accumulated in the lower layers and in the sea meets the cold that arrives in the upper layers of the atmosphere, producing strong storms and downpours, with very intense rainfall concentrated in short periods of time (stationary or quasi-stationary convective nuclei). Depending on the orography, flash floods often occur. As a significant example of an HLID during the period analysed, the tragedy in the coastal town of Sant Llorenç des Cardassar in Majorca on 9 October 2018, which caused 13 fatalities, stands out. An extraordinary convective rainfall event (400 mm in 6 h), unforeseen by most numerical prediction models, generated a devastating flash flood in the urban centre of Sant Llorenç, crossed by a torrent (Ses Planes: 23.4 $km^2$ in drainage area) that reached a peak flow of 305 $m^3/s$. This event far exceeds the extension for a 500-year return period [32,33].

Storm "Gloria", that occurred on 19–20 January 2020, activated the maximum alert level for wind gusts, rain, snow and coastal phenomena (Figure 5) in a large part of the eastern peninsular and in the Balearic Islands [34]. The storm, with heavy rainfall of up to 400 mm/24 h and winds of up to 140 km/h [19], caused 13 fatalities, 3 missing persons and economic damages valued at €71 M (Consorcio de Compensación de Seguros). Gloria set a milestone in the Mediterranean, breaking records of all kinds: snow thickness (86 cm in Vilafranca, Castellón), wave height in the western Mediterranean (8.44 m significant height at the Valencia buoy), strong winds (up to 140 km/h), the maximum daily rainfall collected during the month of January (400 mm/24 h), etc. [35].

On the Spanish Atlantic coast, 30 significant events were recorded during the period analysed, with 9 fatalities. The most significant event occurred on the coast of Galicia,

between 4–7 January 2014, with a sea storm of strong intense waves that triggered the overflowing of rivers and streams and significant flooding on the coast. Three people lost their lives and their bodies were rescued by the Maritime Safety and Rescue Society of Spain.

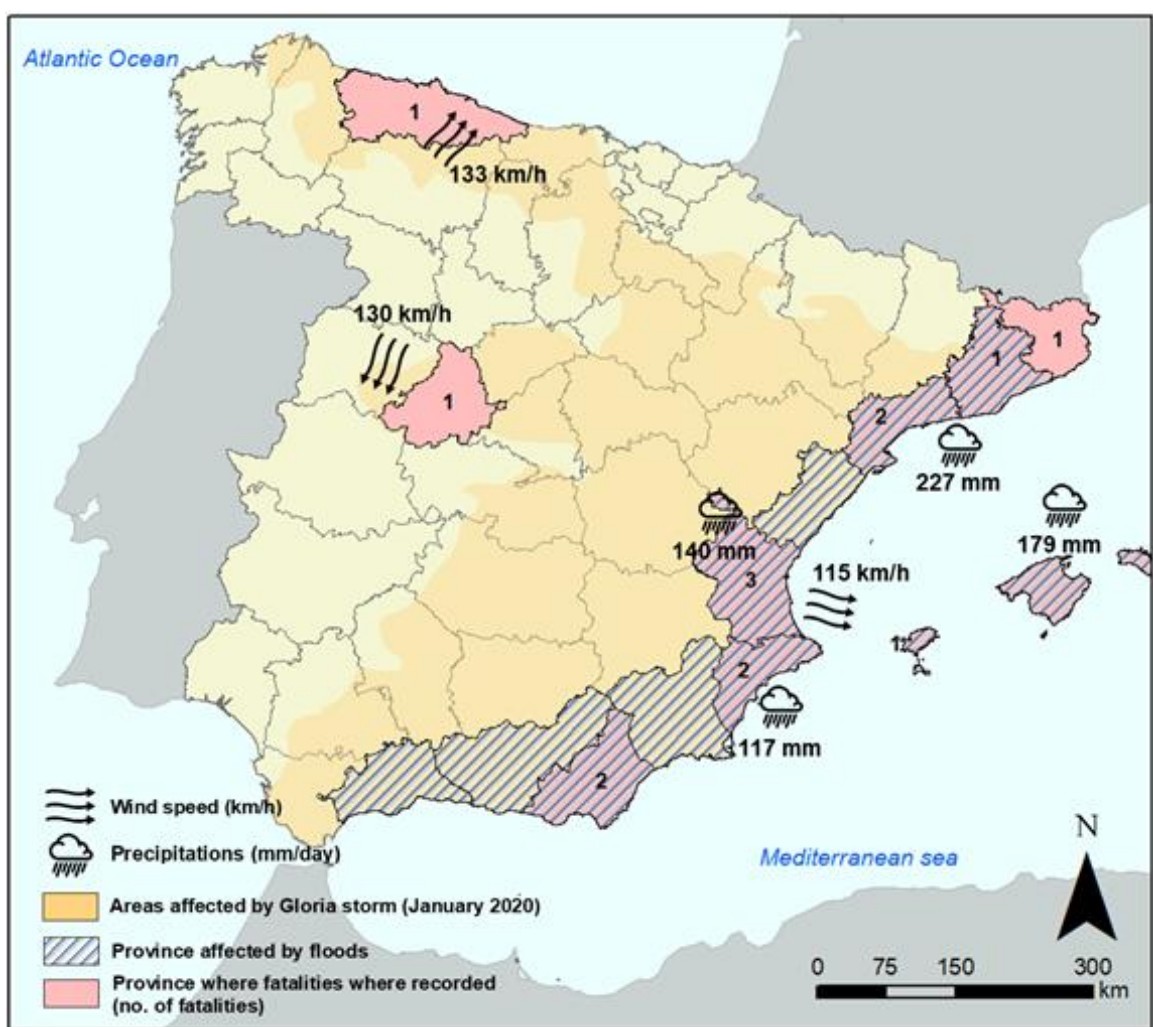

**Figure 5.** Impacts of Storm Gloria (January 2020) in Spain. Fatalities are located by provinces. Gloria set a milestone in the Mediterranean coast of Spain, breaking records of snow thickness, wave thickness and intense rainfall. Data from [36,37].

*4.2. Temporal Data Analysis*

In relation to the temporal distribution (by season) of the events for each of the basins (Mediterranean and Atlantic), the results are shown in Table 3. A difference can be seen between the distribution of events in the two basins. On the Atlantic coast, 71.5% of the events are concentrated during the winter months, while in the Mediterranean basin, the highest concentration of events (58%) occurs in late summer and autumn. It is worth noting that, during the last 5 years of the analysis, an increase of events during the winter has been detected in the Mediterranean basin.

**Table 3.** Distribution, by basin, of the number of extreme weather events recorded for each season of the year.

| Basin | Winter | Spring | Summer | Autumn | Total |
|---|---|---|---|---|---|
| Atlantic | 35 | 4 | 2 | 8 | 49 |
| Mediterranean | 16 | 3 | 9 | 18 | 46 |
| **Total** | 51 | 7 | 11 | 26 | 95 |

Figure 6 shows the evolution over time, for the period analysed, of the number of events recorded, fatalities and economic losses. Regarding the number of events, a slight increase has been observed over the last decade, with 2014 standing out with 20 significant events, followed by 2018 and 2019, with 13 and 12 events, respectively. In relation to fatalities and economic losses, Cyclone Xynthia in France (2010) represents a turning point in both parameters, with the year 2012 also standing out with 26 fatalities in only 4 events, all on the Andalusian Mediterranean coast (13 fatalities in the HLID of 28 September 2012; [38]). In the last years of the analysis (2018, 2019 and early 2020), there was also a slight increase in fatalities (57 victims) and economic costs, with 2019 standing out with almost €563 M in economic losses.

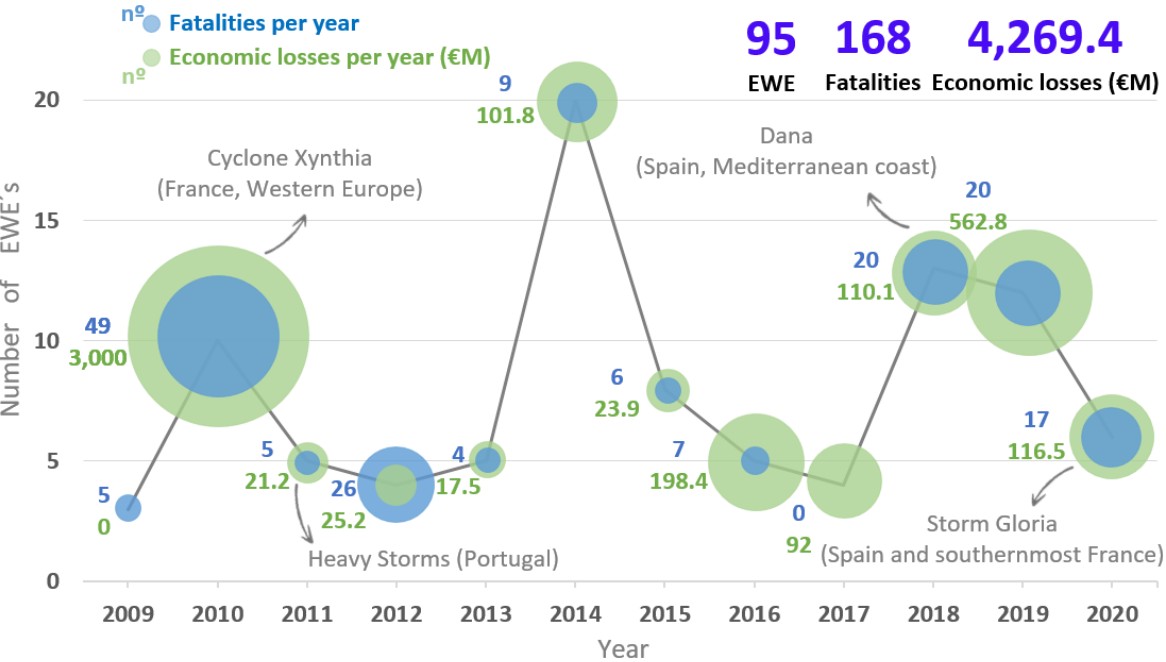

**Figure 6.** Annual distribution of the number of extreme weather events (EWE), fatalities, and economic damage for the study period (2009–2020). For 2020, only the months of January and February were analysed.

Finally, in relation to the causes of death, the main cause has been flood dragging and subsequent drowning (around 60% of the victims with data). The second cause has been in connection to falling trees and/or structural elements (walls, cornices, etc.). The rest of the victims have been due to rock falls and landslides, sea strikes and other, unidentified, causes.

## 5. Discussion

This paper has carried out a basic analysis of the socio-economic impacts caused by 95 extreme weather events that left a clear mark on the southwest coast of Europe during

the period from January 2009 to February 2020. The discussion is organized into three main blocks and a general discussion.

### 5.1. Spatial and Temporal Distribution of the Extreme Weather Events

By country, Spain reports the most events (74 events out of 95) due to its longer coastline (6322 km). Portugal reports 17 events (on 848 km of coastline) and France, 4 (864 km of coastline). By basins, the Atlantic coast, despite having a shorter coastline (3807 km), recorded slightly more events, 49, compared to 46 events on the Mediterranean coast (4227 km). The oceanic climate of the Atlantic coast, with powerful tides, winds and waves, can explain this as well as changes in storminess of this region with extratropical cyclones increasing in the framework of climate change [4].

A different pattern is observed in the Atlantic and Mediterranean basins in terms of the seasonal concentration of events. In the Atlantic basin, winter marine storms predominate, accompanied by strong winds, heavy waves and unusual tides to result in significant localized increases in coastal erosion. In the Mediterranean basin, damaging events are concentrated in late summer and usually during autumn and primarily correspond to HLIDs (episodes of intense rainfall concentrated in short periods of time). This fact coincides with that analysed by Gil-Guirado et al. (2022) on the Spanish Mediterranean coast, where they state that autumn is the season with the highest number of floods and when the most intense and extensive floods are recorded [15]. In this work we also show an increase of winter marine storms in the Mediterranean during the last 5 years of the analysis.

There is a slight upward trend in the number of recorded events, especially during the last decade of the analysis. This result seems to be in line with those obtained in the recent work of Gil-Guirado et al. (2022) [15], that analyses flood events on the Spanish Mediterranean coast and shows an increase in hazard during the period analysed (1960–2015) in terms of the number and extent of events.

### 5.2. Fatalities and Injuries

168 fatalities due to extreme weather events have been confirmed in coastal areas of Southwest Europe during the period analysed. The Spanish Mediterranean coast stands out with 61% of the fatalities. This fact may be due not only to a greater exposure of the population, as it is a preferential tourist destination, but also to the singularity of the events recorded on the Mediterranean coast, primarily HLIDs, that represent sudden events of high rainfall with hardly any margin for response. The Sant Llorenç des Cardassar (Mallorca) event (October 2018) stands out as one of the most tragic cases. Additionally, increases in wave storm duration in Western Mediterranean are primarily responsible for increases in wave storm intensities over the last decades [5], as Storm Gloria (January 2020) occurred on the Spanish Mediterranean coast with 13 fatalities. As concluded by Petrucci et al. (2019) in their analysis of flood fatalities in Europe [12] and in the Euro-Mediterranean region [14], the main cause of mortality identified in this paper is also drowning.

The most tragic event in the series occurred on the French Atlantic coast in February 2010, where Cyclone Xynthia left a trail of 47 deaths and 79 injuries, mostly from the consequences of associated flooding. Xynthia accounted for 72.3% of the fatalities recorded on the Atlantic coast of the study area (65 deaths). The remaining 48 Atlantic events resulted in 18 fatalities, well below the average number of fatalities/event in the Mediterranean strip (103 deaths). The work by Petrucci et al. (2019) on flood fatalities in Europe during the period 1980–2018 does not differentiate between the Atlantic and Mediterranean basin, but it does indicate an increase in fatalities in Mediterranean countries like Greece and Italy, which would be in line with the results of our analysis [12].

There is a slight upward trend in the number of fatalities and injuries during the last three years of the analysis, particularly in the Mediterranean basin. This could be related to a certain increase of extreme weather events during this period and the increased exposure related to tourism and urban development.

*5.3. Economic Impacts*

The economic losses quantified in this paper amount to almost €4000 M during the period analysed. These figures are underestimated for two reasons: (a) there is no reliable information available on the economic impacts of 36 events (almost 38% of the total), and (b) they represent direct costs, mostly for housing and infrastructure repairs, without any analysis of the indirect costs: interruption of commercial activity and tourism, closure of restaurants, hotels and services, loss of profits, etc. Again, Cyclone Xynthia on the Atlantic coast of France (2010) represents the event with the highest recorded economic losses, amounting to €3000 M. Leaving out the economic consequences of Xynthia, the rest of the recorded economic losses are primarily concentrated on the Mediterranean coast (€907.2 M).

In general, results of this study seem to agree with recent studies that report that damage caused by heavy rainfall events and storm surges have increased in Europe during the last decades [6–9]. This is not only related to the effects of climate change, but also to the increased exposure of people, properties and infrastructures to such extreme processes along the coastal fringe. Additionally, the temporal and spatial distribution of events shown here go along with previous research on the increased climatic severity of winters on the European Atlantic coast [4]. This new storm pattern would have important socio-economic repercussions and physical changes in the Atlantic coastal environment. Xynthia (February 2010) is a clear example of this [30]; there are no known coastal flooding events in France with more casualties than Xynthia [27]. On the other hand, a relevant increase in the intensity of storm surges is observed on the Spanish Mediterranean coast [5]; the recent and unusual Storm Gloria (January 2020) was responsible for considerable damage on the Spanish coasts, which requires consideration to manage the risks related to these new scenarios.

This work also shows a greater exposure of the Mediterranean coast of Southwest Europe compared to the Atlantic coast. The Mediterranean coast offers a model of exponential tourist growth, with an enormous urban and infrastructure development during the last decades, which clearly determines a greater vulnerability. Additionally, the Mediterranean coast is less accustomed to dealing with marine storms, not only because of their lower frequency and virulence compared to the Atlantic coast, but also because of a cultural bias of historical traditions. Most of the settlements in the Spanish and French Mediterranean were built far from the coastline to avoid invasions. Portillo et al., (2022) also highlight the greater vulnerability of the Spanish Mediterranean coast to the impacts of climate change. They highlight the vulnerability of small enclaves of ports and marinas that have proliferated enormously along the Mediterranean coast, and which can be severely affected by small changes in significant wave height, slight rises in sea level, etc.

This work has some limitations and weaknesses. Regarding trends observed in the number of extreme meteorological events and socio-economic impacts, the period analysed (11 years and two months) is too short to confirm them. On the other hand, Xynthia (year 2010) has such weight that it makes this type of analysis difficult. Additionally, there are two further limitations: (1) the impossibility of quantifying indirect economic losses, which can multiply the official figures, and (2) the fact that this is a simple analysis conducted by collecting social and economic impacts from official sources. We have identified many information gaps that we have tried to fill with media data. In the case of Portugal, data on economic losses are only available for 3 events, leaving the remaining 14 unquantified. In general, damages are certainly underestimated in the present work.

This work highlights the need to create a continuous monitoring system, at European level, of the impacts generated by adverse meteorological events on the coast and reaffirms the need to extend and standardise this type of work for a sufficiently long period of analysis to allow us to observe significant changes in land use, variations in the stable and tourist population, the functionality of prevention measures already implemented and many other changing parameters that are so characteristic of coastal regions. The coast is not only a highly coveted territory, but also particularly vulnerable to the effects of climate

change. It is the place where marine and terrestrial processes (mainly fluvial) converge, and any slight modification of this equilibrium can have severe impacts. Reliable and standardised information is needed to see that climate change is increasing the frequency and intensity of such coastal events, and that their consequences on the coast are becoming more and more devastating.

## 6. Conclusions

The main conclusions of the present work are:

- During the period spanning from 1 January 2009 and 28 February 2020, 95 extreme weather events were inventoried on the southwest coast of Europe, which comprises 20 coastal regions in south France, continental Portugal and Spain, with a total coastline length of 8034 km.
- The 95 events correspond to meteorologically significant events with a regional impact. They are characterized by heavy precipitation, strong winds, high waves and unusual tides that are likely to trigger chain processes. They also record at least one fatality or injury and economic damage of more than €0.5 M.
- The Atlantic coast (3807 km) recorded slightly more events (49 events) compared to 46 events on the Mediterranean coast (4227 km).
- Andalusia (South Spain), in the Atlantic and Mediterranean basins, is the region with the highest number of events recorded (28). Baleares and Catalonia (Mediterranean) as well as Galicia (Atlantic) rank second in the number of events (12 events each).
- A difference is observed between the temporal-distribution of events in the two basins. While in the Atlantic basin, events are predominantly concentrated during the winter, in the Mediterranean basin the highest concentration of events occurs in late summer and autumn.
- Regarding the number of events, a slight increase has been observed over the last decade of the study period.
- 168 fatalities and 137 injuries due to extreme weather events were confirmed in coastal areas of Southwest Europe during the study period, with flood-drowning the main cause of mortality.
- In terms of fatalities, the Spanish Mediterranean coast stands out with 61% of the total fatalities. It could be related to a greater exposure of the Mediterranean coast and the climate conditions, with many HLIDs episodes of intense rainfall concentrated in short periods of time.
- The total direct economic losses quantified during the study period (134 months) amount to almost €4000 M.
- In the last years of the analysis (2018, 2019 and early 2020), there was a slight increase in fatalities and economic costs. This could be related to increased exposure.
- The most tragic event in the series occurred on the French Atlantic coast in February 2010, where Cyclone Xynthia left a trail of 47 deaths and 79 injuries and economic losses quantified around €3000 M.
- The second most important event was Storm Gloria (January 2020) on the Spanish and French Mediterranean coast. Gloria set a milestone in the Mediterranean, breaking records of snow thickness, wave height and daily rainfall. The storm caused 13 fatalities and direct economic damage valued at €71 M.
- This work highlights a systematic lack of data on the actual impacts caused by extreme weather events on the European coast. Indirect economic costs are not assessed and can presumably overstate the quantified economic impacts.

**Supplementary Materials:** The following supporting information can be downloaded at: https://www.mdpi.com/article/10.3390/app13042640/s1, Databases of EWE during the period 2009–2020 for Spain, Portugal and France.

**Author Contributions:** R.M.M. (IGME) led and designed the work, collected data and write the paper. R.S. (IGME) worked on the methodology, analyzing the data, collecting data from Spain

and drafting the manuscript. A.D.-H., C.R.-C., J.L.-V., P.E., M.M.-C., G.B., J.A.L. (IGME); A.B., O.M. (CTTC); P.M. (ASITEC); A.M., J.P.G., J.M.A., M.O., A.L. (UGR) collaborated in collecting data from Spain, analyzing the data and they critically revised the manuscript. S.P., P.P.S., J.L.Z., E.R., R.A.C.G. and S.C.O. (IGOT) collaborated in collecting data from Portugal, analyzing the data and they critically revised the manuscript. A.V., A.C. and M.G.-B. (CEREMA) collaborated in collecting data from France, analyzing the data and they critically revised the manuscript. All authors have read and agreed to the published version of the manuscript.

**Funding:** This work is funded by the RISKCOAST project (Ref: SOE3/P4/E0868) within the framework of the IV INTERREG SUDOE program.

**Institutional Review Board Statement:** Not applicable.

**Informed Consent Statement:** Not applicable.

**Data Availability Statement:** Not applicable.

**Acknowledgments:** We would like to thank all the French, Portuguese and Spanish institutions that have provided us with data for the development of this work.

**Conflicts of Interest:** The authors declare no conflict of interest.

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
