# Peer review of "Assessment of the Socio-Economic Impacts of Extreme Weather Events on the Coast of Southwest Europe during the Period 2009–2020"

_applsci, doi:10.3390/app13042640_

Round 1
Reviewer 1 Report
The manuscript deals with an interesting topic. However, major revisions are needed before publication.
In detail, to improve the readability of section 3 it would be useful to insert a bulleted list of the phases of the methodology. After this list, the detailed description of each phase, already present in this version of the manuscript, can be shown.
The results are shown aggregated by country. However, the coastal length of the three countries is very different, so it would be useful to relate the results to the coastal length. Furthermore, various extreme events are mentioned but quantitative information is lacking in almost all of them.
In the discussion it would be useful to provide more information about the climatic differences between the Mediterranean Sea and the Atlantic Sea, which cause differences in the characteristics of the perturbations and, therefore, of the extreme events.
Finally, there are numerous typos such as different characters, open but not closed parentheses and the recurring phrase "Error! Reference source not found.".
Author Response
Dear Reviewer,
Please, find attached the cover letter and the revised manuscript. We would like to express our sincere gratitude for your in-depth revision, which unquestionably help us to improve the manuscript.
Yours faithfully,

Reviewer 2 Report
This paper discusses the socio-economic impacts of extreme weather events on the coast of Southwest Europe during the period 2009-2020.It also provides reference information for the government's disaster prevention. There are some suggestions for the authors in the following.
1、Introduction:
Paragraphs 2-4 of the introduction need to be supplemented with references, and the references in paragraph 5 are too general and require some specific conclusions from the references.
2、Characterisation of the study area
Most of the description about the study area in section 2 can be included in the introduction. A brief description and a schematic diagram (Figure 1)of the study area can be combined with the current section 3: data and methodology.
3、Work methodology.
Please change” Work methodology” to “data and methodology” .
In this section, the authors only describe the three types of data collected in general terms, and do not introduce specific research methods. This section should be supplemented with references about data sources and introduce some research methods to be used below.
4、Results
In section 4, the authors mainly introduce the distribution relationship among extreme events number, human impact and economic losses from different regional perspectives. Such an analysis has some significance, but the overall feeling is too simplistic.
Because the title of the paper is “Quantitative assessment…”, the results of the current study do not serve the purpose of quantitative assessment. Maybe, some index should be further defined, and some statistical methods should be used to delve into the relationship between extreme weather events and economic loss and human health.
Author Response
Dear Reviewer,
Please, find attached the cover letter and the revised manuscript.
We would like to express our sincere gratitude for the in-depth revision by you, which unquestionably help us to improve the manuscript.
Yours faithfully,

Reviewer 3 Report
The paper entitled “Quantitative assessment of the socio-economic impacts of extreme weather events on the coast of Southwest Europe during 3 the period 2009-2020” aims to analyse the socio-economic impact of extreme weather events on the coastal area of SW Europe (both the Atlantic and the Mediterranean sides). The manuscript has potential to be published, but needs polishing and English language checking. Some methodological aspects need to be clarified and other (that are missing) added. In the current form, the methodology is vague and the analysis cannot be replicated.
Also, the Introduction needs to be augmented, and the Discussion & Conclusion section needs to be split in 2 distinctive parts. All of these are detailed in the comments inserted throughout the attached pdf. I hope the authors will find them useful and that they will improve the manuscript based on these suggestions.
I appreciated the straight forward writing style and the aesthetic maps and graphs (which increase the attractivity of the paper).

Author Response

(The authors gave the same response as above.)

Round 2
Reviewer 1 Report
The manuscript has been extensively revised, and the authors have satisfactorily responded to the reviewers' comments. Therefore, it is possible to accept the manuscript in present form.
Author Response
Dear Reviewer,
We would like to express our sincere gratitude for your time and dedication.
Many thanks and kind regards,
The Authors

Reviewer 2 Report
Dear editor,
The authors have revised the full text according to the first comments, and have given a reasonable explanation for some of my questions. I agree to accept the article.
Best wishes
Liping Li
Author Response
Dear reviewer,
We would like to express our sincere gratitude for your time and dedication.
Many thanks and kind regards,
The Authors

Reviewer 3 Report
I appreciate the efforts the authors put to improve the paper and congratulate them for the results. The paper is almost good to go to publication, except for the Discussion section, which still needs improvements in order to be at an ISI journal standard.
Please follow the intrusctions provided the last time:
"The structure of the Discussion: - critical interpretation of the results - implication of the results - possible explanation for the results] - comparison of the results with findings of similar papers - contribution of the paper to the scientific literature (Strenghts) - limitations of the paper - future research perspectives."
Right now, Discussion contains just the first 2 parts (critical interpretation + possible explanations). The authors need to compare their findings with previous ones from the literature, regarding the same study area or other study areas around the world. Also, the limitations of the paper should be stated.
Author Response
Dear reviewer,
Please, find attached the new revision of the manuscript taking into account your comments.
We would like to express our sincere gratitude for the in-depth revision by you, as well as for your time and dedication.
Many thanks and kind regards,
The Authors
